# Universal Approximation of Input-Output Maps by Temporal Convolutional Nets

**Joshua Hanson**
University of Illinois
Urbana, IL 61801
jmh4@illinois.edu

**Maxim Raginsky**
University of Illinois
Urbana, IL 61801
maxim@illinois.edu

## Abstract

There has been a recent shift in sequence-to-sequence modeling from recurrent network architectures to convolutional network architectures due to computational advantages in training and operation while still achieving competitive performance. For systems having limited long-term temporal dependencies, the approximation capability of recurrent networks is essentially equivalent to that of temporal convolutional nets (TCNs). We prove that TCNs can approximate a large class of input-output maps having approximately finite memory to arbitrary error tolerance. Furthermore, we derive quantitative approximation rates for deep ReLU TCNs in terms of the width and depth of the network and modulus of continuity of the original input-output map, and apply these results to input-output maps of systems that admit finite-dimensional state-space realizations (i.e., recurrent models).

## 1 Introduction

Until recently, recurrent networks have been considered the de facto standard for modeling input-output maps that transform sequences to sequences. Convolutional network architectures are becoming favorable alternatives for several applications due to reduced computational overhead incurred during both training and regular operation, while often performing as well as or better than recurrent architectures in practice. The computational advantage of convolutional networks follows from the lack of feedback elements, which enables shifted copies of the input sequence to be processed in parallel rather than sequentially [Gehring et al., 2017]. Convolutional architectures have demonstrated exceptional accuracy in sequence modeling tasks that have typically been approached using recurrent architectures, such as machine translation, audio generation, and language modeling [Dauphin et al., 2017, Kalchbrenner et al., 2016, van den Oord et al., 2016, Wu et al., 2016, Gehring et al., 2017, Johnson and Zhang, 2017].

One explanation for this shift is that both convolutional and recurrent architectures are inherently suited to modeling systems with limited long-term dependencies. Recurrent models possess infinite memory (the output at each time is a function of the initial conditions and the entire history of inputs until that time), and thus are strictly more expressive than finite-memory autoregressive models. However, in synthetic stress tests designed to measure the ability to model long-term behavior, recurrent architectures often fail to learn long sequences [Bai et al., 2018]. Furthermore, this unlimited memory property is usually unnecessary, which is supported in theory [Sharan et al., 2018] and in practice [Chelba et al., 2017, Gehring et al., 2017]. In situations where it is only important to learn finite-length sequences, feedforward architectures based on temporal convolutions (temporal convolutional nets, or TCNs) can achieve similar results and even outperform recurrent nets [Dauphin et al., 2017, Yin et al., 2017, Bai et al., 2018].

These results prompt a closer look at the conditions under which convolutional architectures provide better approximation than recurrent architectures. Recent work by Miller and Hardt [2019] has shown

that recurrent models that are exponentially stable (in the sense that the effect of the initial conditions on the output decays exponentially with time) can be efficiently approximated by feedforward models. A key consequence is that exponentially stable recurrent models can be approximated by systems that only consider a finite number of recent values of the input sequence for determining the value of the subsequent output.

However, this notion of stability is inherently tied to a particular state-space realization, and it is not difficult to come up with examples of sequence-to-sequence maps that have both a stable and an unstable state-space realization (e.g., simply by adding unstable states that do not affect the output). This suggests that the question of approximating sequence-to-sequence maps by feedforward convolutional maps should be studied by abstracting away the notion of stability and only requiring that the system output depend appreciably on recent input values and negligibly on input values in the distant past. The formalization of this property was introduced by Sandberg [1991] under the name of *approximately finite memory*, building on earlier work by Boyd and Chua [1985]. Outputs of systems characterized by this property can be approximated by the output of the same system when applied to a truncated version of the input sequence. These systems are naturally suited to be modeled using TCNs, which by construction only operate on values of the input sequence for times within a finite horizon into the past.

In this work, we aim to develop quantitative results for the approximation capability of TCNs for modeling input-output maps that have the properties of causality, time invariance, and approximately finite memory. In Section 2, we introduce the necessary definitions and review the approximately finite memory property due to Sandberg [1991]. Section 3 gives the main result for approximating input-output maps by ReLU TCNs, together with a quantitative result on the equivalence between approximately finite memory and a related notion of *fading memory* [Boyd and Chua, 1985, Park and Sandberg, 1992]. These results are applied in Section 4 to recurrent models that are *incrementally stable* [Tran et al., 2017], i.e., the influence of the initial condition is asymptotically negligible. We show that incrementally stable recurrent models have approximately finite memory, and then use this formalism to derive a generalization of the result of Miller and Hardt [2019]. We provide a comparison in Section 5 to other architectures used for approximating input-output maps. All omitted proofs are provided in the Supplementary Material.

## 2 Input-output maps and approximately finite memory

Let $\mathcal{S}$ denote the set of all real-valued sequences $\boldsymbol{u} = (u_t)_{t \in \mathbb{Z}_+}$, where $\mathbb{Z}_+ := \{0, 1, 2, \ldots\}$. An *input-output map* (or i/o map, for short) is a nonlinear operator $\mathsf{F} : \mathcal{S} \to \mathcal{S}$ that maps an input sequence $\boldsymbol{u} \in \mathcal{S}$ to an output sequence $\boldsymbol{y} = \mathsf{F}\boldsymbol{u} \in \mathcal{S}$. (We are considering real-valued input and output sequences for simplicity; all our results carry over to vector-valued sequences at the expense of additional notation.) We will denote the application and the composition of i/o maps by concatenation. In this paper, we are concerned with i/o maps $\mathsf{F}$ that are:

- *causal* — for any $t \in \mathbb{Z}_+$, $\boldsymbol{u}_{0:t} = \boldsymbol{v}_{0:t}$ implies $(\mathsf{F}\boldsymbol{u})_t = (\mathsf{F}\boldsymbol{v})_t$, where $\boldsymbol{u}_{0:t} := (u_0, \ldots, u_t)$;
- *time-invariant* — for any $k \in \mathbb{Z}_+$,

$$(\mathsf{F}\mathsf{R}^k\boldsymbol{u})_t = \begin{cases} (\mathsf{F}\boldsymbol{u})_{t-k}, & \text{for } t \geq k \\ 0, & \text{for } 0 \leq t < k \end{cases},$$

where $\mathsf{R} : \mathcal{S} \to \mathcal{S}$ is the right shift operator $(\mathsf{R}\boldsymbol{u})_t := u_{t-1}\mathbf{1}_{\{t \geq 1\}}$.

The key notion we will work with is that of *approximately finite memory* [Sandberg, 1991]:

**Definition 2.1.** *An i/o map* $\mathsf{F}$ *has* approximately finite memory *on a set of inputs* $\mathcal{M} \subseteq \mathcal{S}$ *if for any* $\varepsilon > 0$ *there exists* $m \in \mathbb{Z}_+$, *such that*

$$\sup_{\boldsymbol{u} \in \mathcal{M}} \sup_{t \in \mathbb{Z}_+} \left| (\mathsf{F}\boldsymbol{u})_t - (\mathsf{F}\mathsf{W}_{t,m}\boldsymbol{u})_t \right| \leq \varepsilon, \tag{1}$$

*where* $\mathsf{W}_{t,m} : \mathcal{S} \to \mathcal{S}$ *is the* windowing operator $(\mathsf{W}_{t,m}\boldsymbol{u})_\tau := u_\tau \mathbf{1}_{\{\max\{t-m,0\} \leq \tau \leq t\}}$. *We will denote by* $m_{\mathsf{F}}^*(\varepsilon)$ *the smallest* $m \in \mathbb{Z}_+$, *for which* (1) *holds.*

If $m_{\mathsf{F}}^*(0) < \infty$, then we say that $\mathsf{F}$ has *finite memory* on $\mathcal{M}$. If $\mathsf{F}$ is causal and time-invariant, this is equivalent to the existence of an integer $m \in \mathbb{Z}_+$ and a nonlinear functional $f : \mathbb{R}^{m+1} \to \mathbb{R}$, such

that $f(0, \ldots, 0) = 0$ and, for any $\boldsymbol{u} \in \mathcal{M}$ and any $t \in \mathbb{Z}_+$,

$$(\mathsf{F}\boldsymbol{u})_t = f(u_{t-m}, u_{t-m+1}, \ldots, u_t), \tag{2}$$

with the convention that $u_s = 0$ if $s < 0$. In this work, we will focus on the important case when $f$ is a feedforward neural net with rectified linear unit (ReLU) activations $\mathrm{ReLU}(x) := \max\{x, 0\}$. That is, there exist $k$ affine maps $A_i : \mathbb{R}^{d_i} \to \mathbb{R}^{d_{i+1}}$ with $d_1 = m + 1$ and $d_{k+1} = 1$, such that $f$ is given by the composition

$$f = A_k \circ \mathrm{ReLU} \circ A_{k-1} \circ \mathrm{ReLU} \circ \ldots \circ \mathrm{ReLU} \circ A_1,$$

where, for any $r \geq 1$, $\mathrm{ReLU}(x_1, \ldots, x_r) := (\mathrm{ReLU}(x_1), \ldots, \mathrm{ReLU}(x_r))$. Here, $k$ is the depth (number of layers) and $\max\{d_2, \ldots, d_k\}$ is the width (largest number of units in any hidden layer).

**Definition 2.2.** *An i/o map $\mathsf{F}$ is a* ReLU temporal convolutional net *(or ReLU TCN, for short) with context length $m$ if* (2) *holds for some feedforward ReLU neural net $f : \mathbb{R}^{m+1} \to \mathbb{R}$.*

**Remark 2.3.** While such an $\mathsf{F}$ is evidently causal, it is generally not time-invariant unless $f(0, \ldots, 0) = 0$.

## 3  The universal approximation theorem

In this section, we state and prove our main result: any causal and time-invariant i/o map that has approximately finite memory and satisfies an additional continuity condition can be approximated arbitrarily well by a ReLU temporal convolutional net. In what follows, we will consider i/o maps with uniformly bounded inputs, i.e., inputs in the set

$$\mathcal{M}(R) := \{\boldsymbol{u} \in \mathcal{S} : \|\boldsymbol{u}\|_\infty := \sup_{t \in \mathbb{Z}_+} |u_t| \leq R\} \qquad \text{for some } R > 0.$$

For any $t \in \mathbb{Z}_+$ and any $\boldsymbol{u} \in \mathcal{M}(R)$, the finite subsequence $\boldsymbol{u}_{0:t} = (u_0, \ldots, u_t)$ is an element of the cube $[-R, R]^{t+1} \subset \mathbb{R}^{t+1}$; conversely, any vector $\boldsymbol{x} \in [-R, R]^{t+1}$ can be embedded into $\mathcal{M}(R)$ by setting $u_s = x_s \mathbf{1}_{\{0 \leq s \leq t\}}$. To any causal and time-invariant i/o map $\mathsf{F}$ we can associate the nonlinear functional $\tilde{\mathsf{F}}_t : \mathbb{R}^{t+1} \to \mathbb{R}$ defined in the obvious way: for any $\boldsymbol{x} = (x_0, x_1, \ldots, x_t) \in \mathbb{R}^{t+1}$,

$$\tilde{\mathsf{F}}_t(\boldsymbol{x}) := (\mathsf{F}\boldsymbol{u})_t,$$

where $\boldsymbol{u} \in \mathcal{S}$ is any input such that $u_s = x_s$ for $s \in \{0, 1, \ldots, t\}$ (the values of $u_s$ for $s > t$ can be arbitrary by causality). We impose the following assumptions on $\mathsf{F}$:

**Assumption 3.1.** *The i/o map $\mathsf{F}$ has approximately finite memory on $\mathcal{M}(R)$.*

**Assumption 3.2.** *For any $t \in \mathbb{Z}_+$, the functional $\tilde{\mathsf{F}}_t : \mathbb{R}^{t+1} \to \mathbb{R}$ is uniformly continuous on $[-R, R]^{t+1}$ with modulus of continuity*

$$\omega_{t,\mathsf{F}}(\delta) := \sup\left\{|\tilde{\mathsf{F}}_t(\boldsymbol{x}) - \tilde{\mathsf{F}}_t(\boldsymbol{x}')| : \boldsymbol{x}, \boldsymbol{x}' \in [-R, R]^{t+1}, \|\boldsymbol{x} - \boldsymbol{x}'\|_\infty \leq \delta\right\},$$

*and inverse modulus of continuity*

$$\omega_{t,\mathsf{F}}^{-1}(\varepsilon) := \sup\left\{\delta > 0 : \omega_{t,\mathsf{F}}(\delta) \leq \varepsilon\right\}.$$

*where $\|\boldsymbol{x}\|_\infty := \max_{0 \leq i \leq t} |x_i|$ is the $\ell^\infty$ norm on $\mathbb{R}^{t+1}$.*

The following qualitative universal approximation result was obtained by Sandberg [1991]: if a causal and time-invariant i/o map $\mathsf{F}$ satisfies the above two assumptions, then, for any $\varepsilon > 0$, there exists an affine map $A : \mathbb{R}^{m+1} \to \mathbb{R}^d$ and a lattice map $\ell : \mathbb{R}^d \to \mathbb{R}$, such that

$$\sup_{\boldsymbol{u} \in \mathcal{M}(R)} \sup_{t \in \mathbb{Z}_+} \left|(\mathsf{F}\boldsymbol{u})_t - \ell \circ A(\boldsymbol{u}_{t-m:t})\right| < \varepsilon, \tag{3}$$

where we say that a map $\ell : \mathbb{R}^d \to \mathbb{R}$ is a *lattice map* if $\ell(x_0, \ldots, x_{d-1})$ is generated from $x = (x_0, \ldots, x_{d-1})$ by a finite number of min and max operations that do not depend on $x$. Any lattice map can be implemented using ReLU units, so (3) is a ReLU TCN approximation guarantee. Our main result is a quantitative version of Sandberg's theorem:

**Theorem 3.3.** *Let* F *be a causal and time-invariant i/o map satisfying Assumptions 3.1 and 3.2. Then, for any $\varepsilon > 0$ and any $\gamma \in (0, 1)$, there exists a ReLU TCN $\widehat{\mathsf{F}}$ with context length $m = m_{\mathsf{F}}^*(\gamma\varepsilon)$, width $m + 2$, and depth $\left(\frac{O(R)}{\omega_{m,\mathsf{F}}^{-1}((1-\gamma)\varepsilon)}\right)^{m+2}$, such that*

$$\sup_{\boldsymbol{u} \in \mathcal{M}(R)} \|\mathsf{F}\boldsymbol{u} - \widehat{\mathsf{F}}\boldsymbol{u}\|_\infty < \varepsilon. \tag{4}$$

**Remark 3.4.** The role of the additional parameter $\gamma \in (0, 1)$ is to trade off the context length and the depth of the ReLU TCN.

**Remark 3.5.** While the approximating ReLU TCN $\widehat{\mathsf{F}}$ is clearly causal, it may not be time-invariant unless $\widehat{f}(0, \ldots, 0) = 0$, where $\widehat{f}$ is the ReLU net constructed in the proof below.

*Proof.* Let $m = m_{\mathsf{F}}^*(\gamma\varepsilon)$. Since $\tilde{\mathsf{F}}_m : \mathbb{R}^{m+1} \to \mathbb{R}$ is continuous with modulus of continuity $\omega_{m,\mathsf{F}}(\cdot)$, there exists a ReLU net $\widehat{f} : \mathbb{R}^{m+1} \to \mathbb{R}$ of width $m + 2$ and depth $\left(\frac{O(R)}{\omega_{m,\mathsf{F}}^{-1}((1-\gamma)\varepsilon)}\right)^{m+2}$, such that

$$\sup_{\boldsymbol{x} \in [-R,R]^{m+1}} |\tilde{\mathsf{F}}_m(\boldsymbol{x}) - \widehat{f}(\boldsymbol{x})| < (1-\gamma)\varepsilon$$

[Hanin and Sellke, 2018]. Consider the TCN $\widehat{\mathsf{F}}$ defined by $(\widehat{\mathsf{F}}\boldsymbol{u})_t := \widehat{f}(u_{t-m}, \ldots, u_t)$. Fix an input $\boldsymbol{u} \in \mathcal{M}(R)$ and consider two cases:

1) If $t \geq m$, then $\boldsymbol{u}_{t-m:t} = (\mathsf{L}^{t-m}\mathsf{W}_{t,m}\boldsymbol{u})_{0:m}$, where $\mathsf{L} : \mathcal{S} \to \mathcal{S}$ is the left shift operator $(\mathsf{L}\boldsymbol{u})_t := u_{t+1}$. Therefore,

$$(\mathsf{F}\mathsf{W}_{t,m}\boldsymbol{u})_t \overset{(a)}{=} (\mathsf{F}\mathsf{R}^{t-m}\mathsf{L}^{t-m}\mathsf{W}_{t,m}\boldsymbol{u})_t \overset{(b)}{=} (\mathsf{F}\mathsf{L}^{t-m}\mathsf{W}_{t,m}\boldsymbol{u})_m \overset{(c)}{=} \tilde{\mathsf{F}}_m(\boldsymbol{u}_{t-m:t}),$$

where (a) uses the fact that $t \geq m$, (b) is by time invariance of $\mathsf{F}$, and (c) is by the definition of $\tilde{\mathsf{F}}_m$.

2) If $t < m$, then $\boldsymbol{u}_{t-m:t} = (\mathsf{R}^{m-t}\mathsf{W}_{t,m}\boldsymbol{u})_{0:m}$ (recall the convention that, for any $\boldsymbol{v}$, we set $v_s = 0$ whenever $s < 0$). Therefore

$$(\mathsf{F}\mathsf{W}_{t,m}\boldsymbol{u})_t \overset{(a)}{=} (\mathsf{R}^{m-t}\mathsf{F}\mathsf{W}_{t,m}\boldsymbol{u})_m \overset{(b)}{=} (\mathsf{F}\mathsf{R}^{m-t}\mathsf{W}_{t,m}\boldsymbol{u})_m \overset{(c)}{=} \tilde{\mathsf{F}}_m(\boldsymbol{u}_{t-m:t}),$$

where (a) uses the fact that $m > t$, (b) is by time invariance, and (c) is by the definition of $\tilde{\mathsf{F}}_m$.

In either case, the triangle inequality gives

$$|(\mathsf{F}\boldsymbol{u})_t - (\widehat{\mathsf{F}}\boldsymbol{u})_t| \leq |(\mathsf{F}\boldsymbol{u})_t - (\mathsf{F}\mathsf{W}_{t,m}\boldsymbol{u})_t| + |(\mathsf{F}\mathsf{W}_{t,m}\boldsymbol{u})_t - (\widehat{\mathsf{F}}\boldsymbol{u})_t|$$
$$= |(\mathsf{F}\boldsymbol{u})_t - (\mathsf{F}\mathsf{W}_{t,m}\boldsymbol{u})_t| + |\tilde{\mathsf{F}}_m(\boldsymbol{u}_{t-m:t}) - \widehat{f}(\boldsymbol{u}_{t-m:t})|$$
$$< \gamma\varepsilon + (1-\gamma)\varepsilon = \varepsilon.$$

Since this holds for all $t$ and all $\boldsymbol{u}$ with $\|\boldsymbol{u}\|_\infty \leq R$, the result follows. $\square$

## 3.1 The fading memory property

In order to apply Theorem 3.3, we need control on the context length $m_{\mathsf{F}}^*(\cdot)$ and on the modulus of continuity $\omega_{t,\mathsf{F}}(\cdot)$. In general, these quantities are difficult to estimate. However, it was shown by Park and Sandberg [1992] that the property of approximately finite memory is closely related to the notion of *fading memory*, first introduced by Boyd and Chua [1985]. Intuitively, an i/o map F has fading memory if the outputs at any time $t$ due to any two inputs $\boldsymbol{u}$ and $\boldsymbol{v}$ that were close to one another in recent past will also be close.

Let $\mathcal{W}$ denote the subset of $\mathcal{S}$ consisting of all sequences $\boldsymbol{w}$, such that $w_t \in (0, 1]$ for all $t$ and $w_t \downarrow 0$ as $t \to \infty$. We will refer to the elements of $\mathcal{W}$ as *weighting sequences*. Then we have the following definition, due to Park and Sandberg [1992]:

**Definition 3.6.** *We say that an i/o map* F *has* fading memory *on* $\mathcal{M} \subseteq \mathcal{S}$ *with respect to* $\boldsymbol{w} \in \mathcal{W}$ *if for any $\varepsilon > 0$ there exists $\delta > 0$ such that, for all $\boldsymbol{u}, \boldsymbol{v} \in \mathcal{M}$ and all $t \in \mathbb{Z}_+$,*

$$\max_{s \in \{0,\ldots,t\}} w_{t-s}|u_s - v_s| < \delta \quad \Longrightarrow \quad |(\mathsf{F}\boldsymbol{u})_t - (\mathsf{F}\boldsymbol{v})_t| < \varepsilon. \tag{5}$$

The weighting sequence $\boldsymbol{w}$ governs the rate at which the past values of the input are discounted in determining the current output. To capture the best trade-offs in (5), we will also use a $\boldsymbol{w}$-dependent modulus of continuity:

$$\alpha_{\boldsymbol{w},\mathsf{F}}(\delta) := \sup\left\{|(\mathsf{F}\boldsymbol{u})_t - (\mathsf{F}\boldsymbol{v})_t| : t \in \mathbb{Z}_+, \boldsymbol{u}, \boldsymbol{v} \in \mathcal{M}, \max_{s\in\{0,\ldots,t\}} w_{t-s}|u_s - v_s| \le \delta\right\}.$$

It was shown by Park and Sandberg [1992] that an i/o map satisfies Assumptions 3.1 and (3.2) if and only if it has fading memory with respect to some (and hence any) $\boldsymbol{w} \in \mathcal{W}$. The following result provides a quantitative version of this equivalence:

**Proposition 3.7.** *Let* $\mathsf{F}$ *be an i/o map.*

1. *If* $\mathsf{F}$ *satisfies Assumptions 3.1 and 3.2, then it has fading memory on* $\mathcal{M}$ *with respect to any weighting sequence* $\boldsymbol{w} \in \mathcal{W}$, *and*

$$\alpha_{\boldsymbol{w},\mathsf{F}}^{-1}(\varepsilon) \ge w_{m_{\mathsf{F}}^*(\varepsilon/3)} \omega_{m_{\mathsf{F}}^*(\varepsilon/3),\mathsf{F}}^{-1}(\varepsilon/3). \tag{6}$$

2. *If* $\mathsf{F}$ *has fading memory on* $\mathcal{M}(R)$ *with respect to some* $\boldsymbol{w} \in \mathcal{W}$, *then it has satisfies Assumptions 3.1 and 3.2, and*

$$m_{\mathsf{F}}^*(\varepsilon; R) \le \inf\left\{m \in \mathbb{Z}_+ : w_m \le \frac{\alpha_{\boldsymbol{w},\mathsf{F}}^{-1}(\varepsilon)}{R}\right\} \qquad and \qquad \omega_{t,\mathsf{F}}(\delta) \le \alpha_{\boldsymbol{w},\mathsf{F}}(\delta). \tag{7}$$

## 4 Recurrent systems

So far, we have considered arbitrary i/o maps $\mathsf{F} : \mathcal{S} \to \mathcal{S}$. However, many such maps admit *state-space realizations* [Sontag, 1998] — there exist a state transition map $f : \mathbb{R}^n \times \mathbb{R} \to \mathbb{R}^n$, an output map $g : \mathbb{R}^n \to \mathbb{R}$, and an initial condition $\xi \in \mathbb{R}^n$, such that the output sequence $\boldsymbol{y} = \mathsf{F}\boldsymbol{u}$ is detemined recursively by

$$x_{t+1} = f(x_t, u_t) \tag{8a}$$
$$y_t = g(x_t) \tag{8b}$$

with $x_0 = \xi$. The i/o map $\mathsf{F}$ realized in this way is evidently causal, and it is time-invariant if $f(\xi, 0) = \xi$ and $g(\xi) = 0$. In this section, we will identify the conditions under which recurrent models satisfy Assumptions 3.1 and 3.2. Along the way, we will derive the approximation results of Miller and Hardt [2019] as a special case.

### 4.1 Approximately finite memory and incremental stability

Consider the system in (8). Given any input $\boldsymbol{u} \in \mathcal{S}$, any $\xi \in \mathbb{R}^n$, and any $s, t \in \mathbb{Z}_+$ with $t \ge s$, we denote by $\varphi_{s,t}^{\boldsymbol{u}}(\xi)$ the state at time $t$ when $x_s = \xi$. Let $\mathcal{M}$ be a subset of $\mathcal{S}$. We say that $\mathbb{X} \subseteq \mathbb{R}^n$ is a *positively invariant set* of (8) for inputs in $\mathcal{M}$ if, for all $\xi \in \mathbb{X}$, all $\boldsymbol{u} \in \mathcal{M}$, and all $0 \le s \le t$, $\varphi_{s,t}^{\boldsymbol{u}}(\xi) \in \mathbb{X}$. We will be interested in systems with the following property [Tran et al., 2017]:

**Definition 4.1.** *The system* (8) *is* uniformly asymptotically incrementally stable *for inputs in* $\mathcal{M}$ *on a positively invariant set* $\mathbb{X}$ *if there exists a function* $\beta : \mathbb{R}_+ \times \mathbb{R}_+ \to \mathbb{R}_+$ *of class* $\mathcal{KL}$[1], *such that the inequality*

$$\|\varphi_{s,t}^{\boldsymbol{u}}(\xi) - \varphi_{s,t}^{\boldsymbol{u}}(\xi')\| \le \beta(\|\xi - \xi'\|, t - s) \tag{9}$$

*holds for all inputs* $\boldsymbol{u} \in \mathcal{M}$, *all initial conditions* $\xi, \xi' \in \mathbb{X}$, *and all* $0 \le s \le t$, *where* $\|\cdot\|$ *is the* $\ell^2$ *norm on* $\mathbb{R}^n$.

In other words, a system is incrementally stable if the influence of any initial condition in $\mathbb{X}$ on the state trajectory is asymptotically negligible. A key consequence is the following estimate:

**Proposition 4.2.** *Let $\boldsymbol{u}, \tilde{\boldsymbol{u}}$ be two input sequences in $\mathcal{M}$. Then, for any $\xi \in \mathbb{X}$ and any $t \in \mathbb{Z}_+$,*

$$\|\varphi_{0,t}^{\boldsymbol{u}}(\xi) - \varphi_{0,t}^{\tilde{\boldsymbol{u}}}(\xi)\| \leq \sum_{s=0}^{t-1} \beta\left(\|f(\tilde{x}_s, u_s) - f(\tilde{x}_s, \tilde{u}_s)\|, t - s - 1\right), \tag{10}$$

*where $x_s$ and $\tilde{x}_s$ denote the states at time $s$ due to inputs $\boldsymbol{u}$ and $\tilde{\boldsymbol{u}}$, respectively, with $x_0 = \tilde{x}_0 = \xi$.*

Consider a state-space model (8) with a positively invariant set $\mathbb{X}$, with the following assumptions:

**Assumption 4.3.** *The state transition map $f(x, u)$ is $L_f$-Lipschitz in $u$ for all $x \in \mathbb{X}$ and the output map $g(x)$ is $L_g$-Lipschitz in $x \in \mathbb{X}$.*

**Assumption 4.4.** *For any initial condition $\xi \in \mathbb{X}$ there exists a compact set $\mathbb{S}_\xi \subseteq \mathbb{X}$ such that $\varphi_{0,t}^{\boldsymbol{u}}(\xi) \in \mathbb{S}_\xi$ for all $\boldsymbol{u} \in \mathcal{M}(R)$ and all $t \in \mathbb{Z}_+$.*

**Assumption 4.5.** *The system (8) is uniformly asymptotically incrementally stable on $\mathbb{X}$ for inputs in $\mathcal{M}(R)$, and the function $\beta$ in (9) satisfies the summability condition*

$$\sum_{t \in \mathbb{Z}_+} \beta(C, t) < \infty \tag{11}$$

*for any $C \geq 0$. (For example, if $\beta(C, k) = Ck^{-\alpha}$ for some $\alpha > 1$, then this condition is satisfied.)*

We are now in position to prove the main result of this section:

**Theorem 4.6.** *Suppose that Assumptions 4.3–4.5 are satisfied. Then the i/o map $\mathsf{F}$ of the system (8) satisfies Assumptions 3.1 and 3.2 with*

$$m_{\mathsf{F}}^*(\varepsilon) \leq \min\left\{m \in \mathbb{Z}_+ : \sum_{k \geq m} \beta(\mathrm{diam}(\mathbb{S}_\xi), k) < \varepsilon/L_g\right\} \tag{12}$$

*and*

$$\omega_{t,\mathsf{F}}(\delta) \leq L_g \sum_{s=0}^{t-1} \beta(L_f \delta, s), \qquad \forall t \in \mathbb{Z}_+. \tag{13}$$

*Proof.* Fix some $t, m \in \mathbb{Z}_+$. For an arbitrary input $\boldsymbol{u} \in \mathcal{M}(R)$, let $\tilde{\boldsymbol{u}} = \mathsf{W}_{t,m}\boldsymbol{u}$, where we may assume without loss of generality that $t \geq m$. Then $\tilde{u}_s = u_s \mathbf{1}_{\{t-m \leq s \leq t\}}$, and therefore

$$\sum_{s=0}^{t-1} \beta\left(\|f(\tilde{x}_s, u_s) - f(\tilde{x}_s, \tilde{u}_s)\|, t - s - 1\right) = \sum_{s=0}^{t-m-1} \beta\left(\|f(\tilde{x}_s, u_s) - f(\tilde{x}_s, 0)\|, t - s - 1\right)$$

$$\leq \sum_{s=0}^{t-m-1} \beta(\mathrm{diam}(\mathbb{S}_\xi), t - s - 1)$$

$$\leq \sum_{s=m}^{\infty} \beta(\mathrm{diam}(\mathbb{S}_\xi), s). \tag{14}$$

By the summability condition (11), the summation in (14) converges to 0 as $m \uparrow \infty$. Thus, if we choose $m$ so that the right-hand side of (14) is smaller than $\varepsilon/L_g$, it follows from Proposition 4.2 that

$$|(\mathsf{F}\boldsymbol{u})_t - (\mathsf{F}\mathsf{W}_{t,m}\boldsymbol{u})_t| = |g(\varphi_{0,t}^{\boldsymbol{u}}(\xi)) - g(\varphi_{0,t}^{\tilde{\boldsymbol{u}}}(\xi))| \leq L_g\|\varphi_{0,t}^{\boldsymbol{u}}(\xi) - \varphi_{0,t}^{\tilde{\boldsymbol{u}}}(\xi)\| < \varepsilon.$$

This proves (12). Now fix any two $\boldsymbol{u}, \tilde{\boldsymbol{u}} \in \mathcal{M}(R)$ with $\|\boldsymbol{u}_{0:t} - \tilde{\boldsymbol{u}}_{0:t}\|_\infty < \delta$. Then $\max_{0 \leq s \leq t} \|f(x, u_s) - f(x, \tilde{u}_s)\| \leq L_f \delta$ for all $x \in \mathbb{X}$, so Proposition 4.2 gives

$$|\tilde{\mathsf{F}}_t(\boldsymbol{u}_{0:t}) - \tilde{\mathsf{F}}_t(\tilde{\boldsymbol{u}}_{0:t})| = |g(\varphi_{0,t}^{\boldsymbol{u}}(\xi)) - g(\varphi_{0,t}^{\tilde{\boldsymbol{u}}}(\xi))|$$

$$\leq L_g\|\varphi_{0,t}^{\boldsymbol{u}}(\xi) - \varphi_{0,t}^{\tilde{\boldsymbol{u}}}(\xi)\|$$

$$\leq L_g \sum_{s=0}^{t-1} \beta(L_f \delta, s),$$

which proves (13). $\qquad\square$

## 4.2 Exponential incremental stability and the Demidovich criterion

Miller and Hardt [2019] consider the case of contracting systems: there exists some $\lambda \in (0, 1)$ and a set $\mathbb{U} \subseteq \mathbb{R}^m$, such that

$$\|f(x, u) - f(x', u)\| \le \lambda \|x - x'\| \tag{15}$$

for all $x, x' \in \mathbb{R}^n$ and all $u \in \mathbb{U}$. Such a system is *uniformly exponentially incrementally stable* on any positively invariant set $\mathbb{X}$, with $\beta(C, t) = C\lambda^t$. In this section, we obtain their result as a special case of a more general stability criterion, known in the literature on nonlinear system stability as the *Demidovich criterion* [Pavlov et al., 2006]. The following result is a simplified version of a more general result of Tran et al. [2017]:

**Proposition 4.7** (the discrete-time Demidovich criterion). *Consider the recurrent system* (8) *with a convex positively invariant set $\mathbb{X}$, where the state transition map $f(x, u)$ is differentiable in $x$ for any $u \in \mathbb{U}$. Suppose that there exists a symmetric positive definite matrix $P$ and a constant $\mu \in (0, 1)$, such that*

$$\frac{\partial}{\partial x} f(x, u)^\top P \frac{\partial}{\partial x} f(x, u) - \mu P \preceq 0 \tag{16}$$

*for all $x \in \mathbb{X}$ and all $u \in \mathbb{U}$, where $\frac{\partial}{\partial x} f(x, u)$ is the Jacobian of $f(\cdot, u)$ with respect to $x$. Then the system* (8) *is uniformly exponentially incrementally stable with $\beta(C, t) = \sqrt{\kappa(P)} C \mu^{t/2}$, where $\kappa(P)$ is the condition number of $P$.*

*Proof.* Fix any $u \in \mathbb{U}$ and $\xi, \xi' \in \mathbb{X}$, and define the function $\Phi : [0, 1] \to \mathbb{R}$ by

$$\Phi(s) := (f(\xi, u) - f(\xi', u))^\top P f(s\xi + (1-s)\xi', u).$$

Then

$$\Phi(1) - \Phi(0) = (f(\xi, u) - f(\xi', u))^\top P(f(\xi, u) - f(\xi', u)). \tag{17}$$

By the mean-value theorem, there exists some $\bar{s} \in [0, 1]$, such that

$$\Phi(1) - \Phi(0) = \frac{\mathrm{d}}{\mathrm{d}s}\Phi(s)\Big|_{s=\bar{s}} = (f(\xi, u) - f(\xi', u))^\top P \frac{\partial}{\partial x} f(\bar{\xi}, u)(\xi - \xi'), \tag{18}$$

where $\bar{\xi} = \bar{s}\xi + (1-\bar{s})\xi' \in \mathbb{X}$, since $\mathbb{X}$ is convex. From (16), (17), and (18) it follows that

$$(f(\xi, u) - f(\xi', u))^\top P(f(\xi, u) - f(\xi', u))$$
$$\le (\xi - \xi')^\top \frac{\partial}{\partial x} f(\bar{\xi}, u)^\top P \frac{\partial}{\partial x} f(\bar{\xi}, u)(\xi - \xi')$$
$$\le \mu(\xi - \xi')^\top P(\xi - \xi').$$

Define the function $V : \mathbb{X} \times \mathbb{X} \to \mathbb{R}_+$ by $V(\xi, \xi') := (\xi - \xi')^\top P(\xi - \xi')$. From the above estimate, it follows that $V$ is a *Lyapunov function* for the dynamics, i.e., for any $u \in \mathbb{U}$ and $\xi, \xi' \in \mathbb{X}$,

$$V(f(\xi, u), f(\xi', u)) \le \mu V(\xi, \xi'). \tag{19}$$

Consequently, for any input $\boldsymbol{u}$ with $u_t \in \mathbb{U}$ for all $t$ and any $\xi, \xi' \in \mathbb{X}$,

$$V(\varphi_{0,t+1}^{\boldsymbol{u}}(\xi), \varphi_{0,t+1}^{\boldsymbol{u}}(\xi')) = V(f(\varphi_{0,t}^{\boldsymbol{u}}(\xi), u_t), f(\varphi_{0,t}^{\boldsymbol{u}}(\xi'), u_t))$$
$$\le \mu V(\varphi_{0,t}^{\boldsymbol{u}}(\xi), \varphi_{0,t}^{\boldsymbol{u}}(\xi')).$$

Iterating, we obtain the inequality $V(\varphi_{0,t}^{\boldsymbol{u}}(\xi), \varphi_{0,t}^{\boldsymbol{u}}(\xi')) \le \mu^t V(\xi, \xi')$. Finally, since $P \succ 0$,

$$\|\varphi_{0,t}^{\boldsymbol{u}}(\xi) - \varphi_{0,t}^{\boldsymbol{u}}(\xi)\|^2 \le \frac{\lambda_{\max}(P)}{\lambda_{\min}(P)}\mu^t\|\xi - \xi'\|^2 = \kappa(P)\|\xi - \xi'\|^2 \mu^t,$$

and the proof is complete. $\qquad\square$

**Theorem 4.8.** *Suppose the system* (8) *satisfies Assumption 4.3 and the Demidovich criterion with $\mathbb{U} = [-R, R]$, its positively invariant set $\mathbb{X}$ contains 0, and $f(0, 0) = 0$. Then its i/o map $\mathsf{F}$ with zero initial condition $x_0 = 0$ satisfies Assumptions 3.1 and 3.2 with*

$$m_{\mathsf{F}}^*(\varepsilon) \le \frac{2\log\left(\frac{2\kappa(P)L_f L_g R}{(1-\sqrt{\mu})^2 \varepsilon}\right)}{\log\frac{1}{\mu}} \qquad and \qquad \omega_{t,\mathsf{F}}(\delta) \le \frac{\sqrt{\kappa(P)}L_f L_g \delta}{1 - \sqrt{\mu}}. \tag{20}$$

*Proof.* Since $P$ is symmetric and positive definite, $\|x\|_P := \sqrt{x^\top P x}$ is a norm on $\mathbb{R}^n$ with $\lambda_{\min}(P)\|\cdot\|^2 \le \|\cdot\|_P^2 \le \lambda_{\max}(P)\|\cdot\|^2$. Then, for all $\xi \in \mathbb{X}$, $\boldsymbol{u} \in \mathcal{M}(R)$, and $t$,

$$
\begin{aligned}
\|\varphi_{0,t+1}^{\boldsymbol{u}}(\xi)\|_P &= \|f(\varphi_{0,t}^{\boldsymbol{u}}(\xi), u_t)\|_P \\
&\le \|f(\varphi_{0,t}^{\boldsymbol{u}}(\xi), u_t) - f(0, u_t)\|_P + \|f(0, u_t) - f(0,0)\|_P \\
&\le \sqrt{\mu}\|\varphi_{0,t}^{\boldsymbol{u}}(\xi)\|_P + \sqrt{\lambda_{\max}(P)}L_f R,
\end{aligned}
$$

where we have used the Lyapunov bound (19). Unrolling the recursion gives the estimate

$$
\sup_{t \in \mathbb{Z}_+} \sup_{\boldsymbol{u} \in \mathcal{M}(R)} \|\varphi_{0,t}^{\boldsymbol{u}}(\xi)\|_P \le \sqrt{\mu}\|\xi\|_P + \frac{\sqrt{\lambda_{\max}(P)}L_f R}{1 - \sqrt{\mu}}.
$$

Thus, Assumption 4.4 is satisfied, where $\mathbb{S}_\xi$ is the ball of $\ell^2$-radius $\sqrt{\kappa(P)}\left(\|\xi\| + \frac{L_f R}{1-\sqrt{\mu}}\right)$ centered at 0. Assumption 4.5 is also satisfied by Proposition 4.7. The estimates in (20) follow from Theorem 4.6. $\qquad\square$

The following result now follows as a direct consequence of Theorems 3.3 and 4.8:

**Corollary 4.9.** *If the system* (8) *satisfies the conditions of Theorem 4.8, then its i/o map* $\mathsf{F}$ *with zero initial condition can be $\varepsilon$-approximated in the sense of Theorem 3.3 by a ReLU TCN $\widehat{\mathsf{F}}$ with width* $\mathrm{polylog}(\frac{1}{\varepsilon})$ *and depth* $\mathrm{quasipoly}(\frac{1}{\varepsilon})$.[2]

### 4.3 Contractivity vs. the Demidovich criterion

If the contractivity condition (15) holds and $f(x,u)$ is differentiable in $x$, then the Demidovich criterion is satisfied with $P = I_n$ and $\mu = \lambda^2$. In that case, we immediately obtain the exponential estimate $\beta(C,t) \le C\lambda^t$. However, the Demidovich criterion covers a wider class of nonlinear systems. As an example, consider a discrete-time nonlinear system of *Lur'e type* (cf. Sandberg and Xu [1993], Kim and Braatz [2014], Sarkans and Logemann [2016] and references therein):

$$
\begin{aligned}
x_{t+1} &= Ax_t + B\psi(u_t - y_t) && \text{(21a)} \\
y_t &= Cx_t && \text{(21b)}
\end{aligned}
$$

Here, the state $x_t$ is $n$-dimensional while the input $u_t$ and the output $y_t$ are scalar, so $A \in \mathbb{R}^{n \times n}$, $B \in \mathbb{R}^{n \times 1}$, and $C \in \mathbb{R}^{1 \times n}$. The map $\psi : \mathbb{R} \to \mathbb{R}$ is a fixed differentiable nonlinearity. The system in (21) has the form (8) with $f(x,u) = Ax + B\psi(u - Cx)$ and $g(x) = Cx$, and can be realized as the negative feedback interconnection of the discrete-time linear system

$$
\begin{aligned}
x_{t+1} &= Ax_t + Bv_t && \text{(22a)} \\
y_t &= Cx_t && \text{(22b)}
\end{aligned}
$$

and the nonlinear element $\psi$ using the feedback law $v_t = \psi(u_t - y_t)$. We make the following assumptions (see, e.g., Sontag [1998] for the requisite control-theoretic background):

**Assumption 4.10.** *The nonlinearity $\psi : \mathbb{R} \to \mathbb{R}$ satisfies $\psi(0) = 0$, and there exist real numbers $-\infty < a \le b < \infty$ such that $a \le \psi'(\cdot) \le b$.*

**Assumption 4.11.** *$A$ is a* Schur *matrix, i.e., its spectral radius $\rho(A)$ is strictly smaller than 1; the pair $(A,B)$ is* controllable, *i.e., the $n \times n$ matrix $[B \mid AB \mid \ldots \mid A^{n-1}B]$ has rank $n$; and the pair $(A,C)$ is* observable, *i.e., the $n \times n$ matrix $[C^\top \mid A^\top C^\top \mid \ldots \mid (A^\top)^{n-1}C^\top]$ has rank $n$.*

**Assumption 4.12.** *Let $\mathbb{T} := \{z \in \mathbb{C} : |z| = 1\}$ denote the unit circle in the complex plane. The rational function $G(z) := C(zI_n - A)^{-1}B$ satisfies*

$$
\|G\|_{\mathcal{H}_\infty(\mathbb{T})} := \sup_{z \in \mathbb{T}} |G(z)| < \gamma^{-1} \tag{23}
$$

*for some $\gamma > 0$ such that $r^2 \le \gamma^2$ for all $a \le r \le b$.*

**Remark 4.13.** Assumption 4.10 imposes a *slope condition* on $\psi$ and is standard in the analysis of Lur'e systems [Tsypkin, 1964, Sandberg, 1991, Kim and Braatz, 2014]. The function $G(z)$ is the *transfer function* of the linear system (22). Assumption 4.11 states that the triple $(A, B, C)$ is a *minimal realization* of $G$. The quantity $\|G\|_{\mathcal{H}_\infty(\mathbb{T})}$ appearing in Eq. (23) in Assumption 4.12 is the $\mathcal{H}_\infty$-*norm* of $G$ on the unit circle in the complex plane. Assumptions 4.11 and 4.12 are also common and are in the spirit of the well-known *circle criterion* [Tsypkin, 1964, Sandberg and Xu, 1993].

With these preliminaries out of the way, we have the following:

**Proposition 4.14.** *Suppose that the system* (21) *satisfies Assumptions 4.10–4.12. Then it satisfies the discrete-time Demidovich criterion with* $\mathbb{X} = \mathbb{R}^n$ *and* $\mathbb{U} = \mathbb{R}$, *and moreover* $\mu > \rho(A)^2$.

The crucial ingredient in the proof is the Discrete-Time Bounded-Real Lemma [Vaidyanathan, 1985], which guarantees the existence of the matrix $P$ appearing in the Demidovich criterion. The main takeaway here is that the function $f(x, u) = Ax + B\psi(u - Cx)$ need not be contractive (i.e., it may be the case that $P \neq I_n$), but it will be contractive in the $\|\cdot\|_P$ norm.

## 5  Comparison of architectures

So far, we have shown that any i/o map $\mathsf{F}$ with approximately finite memory can be approximated by a ReLU temporal convolutional net. We have also considered recurrent models and shown that any incrementally stable recurrent model has approximately finite memory and can therefore be approximated by a ReLU TCN. As far as their approximation capabilities are concerned, both recurrent models and autoregressive models like TCNs are equivalent, since any finite-memory i/o map of the form (2) admits the state-space realization

$$x^1_{t+1} = x^2_t, x^2_{t+1} = x^3_t, \ldots, x^{m-1}_{t+1} = x^m_t, x^m_{t+1} = u_t$$
$$y_t = f(x^1_t, x^2_t, \ldots, x^m_t, u_t)$$

of the tapped delay line type, with zero initial condition $(x^1_0, \ldots, x^m_0) = (0, \ldots, 0)$. (Compared to (8), we are allowing a direct 'feedthrough' connection from the input $u_t$ to the output $y_t$.) The advantage of autoregressive models like TCNs shows up during training and regular operation, since shifted copies of the input sequence can be efficiently processed in parallel rather than sequentially.

Another point worth mentioning is that, while the construction in the proof of Theorem 3.3 makes use of ReLU nets as a universal function approximator, any other family of universal approximators can be used instead, for example, multivariate polynomials or rational functions. In fact, if one uses multivariate polynomials to approximate the functionals $\tilde{\mathsf{F}}_t$, the resulting family of i/o maps is known as the (discrete-time) finite Volterra series [Boyd and Chua, 1985], and has been used widely in the analysis of nonlinear systems. However, TCNs generally provide a more parsimonious representation. To see this, consider the following (admittedly contrived) example of an i/o map:

$$(\mathsf{F}\boldsymbol{u})_t = \mathrm{ReLU}\left( \sum_{s=0}^{\infty} h_s u_{t-s} \right), \tag{24}$$

where the filter coefficients $h_t$ have the exponential decay property $|h_t| \leq C\lambda^t$ for some $C > 0$ and $\lambda \in (0, 1)$. It is not hard to show that $\mathsf{F}$ has exponentially fading memory, and a very simple $\varepsilon$-approximation by a TCN is obtained by zeroing out all of the filter coefficients $h_s, s > m \sim \log(\frac{1}{\varepsilon})$:

$$(\widehat{\mathsf{F}}\boldsymbol{u})_t = \mathrm{ReLU}\left( \sum_{s=0}^{m} h_s u_{t-s} \right).$$

However, any $\varepsilon$-approximation for $\mathsf{F}$ using Volterra series would need $\mathrm{poly}(\frac{1}{\varepsilon})$ terms, since the best polynomial $\varepsilon$-approximation of the ReLU on any compact interval has degree $\Omega(\frac{1}{\varepsilon})$ [DeVore and Lorentz, 1993, Chap. 9, Thm. 3.3]. On the other hand, if we consider an i/o map of the form (24), but with a degree-$d$ univariate polynomial instead of ReLU, then we can $\varepsilon$-approximate it with a TCN of depth $O(d + \log \frac{d}{\varepsilon})$ and $O(d \log \frac{d}{\varepsilon})$ units [Liang and Srikant, 2017].

**Acknowledgments**

This work was supported in part by the National Science Foundation under the Center for Advanced Electronics through Machine Learning (CAEML) I/UCRC award no. CNS-16-24811.

## Footnotes

[1]A function $\beta : \mathbb{R}_+ \times \mathbb{R}_+ \to \mathbb{R}_+$ is of class $\mathcal{KL}$ if it is continuous and strictly increasing in its first argument, continuous and strictly decreasing in its second argument, $\beta(0, t) = 0$ for any $t$, and $\lim_{t\to\infty} \beta(r, t) = 0$ for any $r$ [Sontag, 1998].

[2]We say that a given quantity $N$ has *quasipolynomial growth* in $1/\varepsilon$, and write $N \le \mathrm{quasipoly}(1/\varepsilon)$, if $N = O(\exp(\mathrm{polylog}(\frac{1}{\varepsilon})))$.

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
