[Supplementary Material · approximately_finite_supplementary.pdf]

# A   Omitted proofs

*Proof of Proposition 3.7.* Suppose $\mathsf{F}$ satisfies Assumptions 3.1 and 3.2. Fix some $\varepsilon > 0$ and let $m = m_{\mathsf{F}}^*(\varepsilon/3)$ and $\delta = w_m \omega_{m,\mathsf{F}}^{-1}(\varepsilon/3)$. Now fix some $t \in \mathbb{Z}_+$ and consider any two $\boldsymbol{u}, \boldsymbol{v} \in \mathcal{M}(R)$ such that

$$\max_{s \in \{0,\ldots,t\}} w_{t-s} |u_s - v_s| < \delta. \tag{A.1}$$

Using the same reasoning as in the proof of Theorem 3.3, we can write $(\mathsf{FW}_{t,m}\boldsymbol{u})_t = \tilde{\mathsf{F}}_m(\boldsymbol{u}_{t-m:t})$ and $(\mathsf{FW}_{t,m}\boldsymbol{v})_t = \tilde{\mathsf{F}}_m(\boldsymbol{v}_{t-m:t})$, where, as before, we set $u_s = v_s = 0$ for $s < 0$. From the monotonicity of $\boldsymbol{w}$ and (A.1) it follows that

$$\|\boldsymbol{u}_{t-m:t} - \boldsymbol{v}_{t-m:t}\|_\infty \le \frac{1}{w_m} \max_{s \in \{t-m,\ldots,t\}} w_{t-s}|u_s - v_s| < \omega_{m,\mathsf{F}}^{-1}(\varepsilon/3),$$

which implies that

$$|(\mathsf{FW}_{t,m}\boldsymbol{u})_t - (\mathsf{FW}_{t,m}\boldsymbol{v})_t| = |\tilde{\mathsf{F}}_m(\boldsymbol{u}_{t-m:t}) - \tilde{\mathsf{F}}_m(\boldsymbol{v}_{t-m:t})| < \varepsilon/3.$$

Altogether, we see that (A.1) implies that

$$|(\mathsf{F}\boldsymbol{u})_t - (\mathsf{F}\boldsymbol{v})_t| \le |(\mathsf{F}\boldsymbol{u})_t - (\mathsf{FW}_{t,m}\boldsymbol{u})_t| + |(\mathsf{FW}_{t,m}\boldsymbol{u})_t - (\mathsf{FW}_{t,m}\boldsymbol{v})_t| + |(\mathsf{F}\boldsymbol{v})_t - (\mathsf{FW}_{t,m}\boldsymbol{v})_t|$$
$$< \varepsilon/3 + \varepsilon/3 + \varepsilon/3 = \varepsilon,$$

which leads to (6).

Now suppose that $\mathsf{F}$ has fading memory w.r.t. $\boldsymbol{w}$. Given $\varepsilon > 0$, let $\delta = \alpha_{\boldsymbol{w},\mathsf{F}}^{-1}(\varepsilon)$ and choose any $m \in \mathbb{Z}_+$, such that $w_m < \delta/R$. If $t < m$, then $\boldsymbol{u}_{0:t} = (\mathsf{W}_{t,m}\boldsymbol{u})_{0:t}$, and thus $(\mathsf{F}\boldsymbol{u})_t = (\mathsf{FW}_{t,m}\boldsymbol{u})_t$. On the other hand, if $t \ge m$, then, for any $\boldsymbol{u} \in \mathcal{M}(R)$,

$$\max_{s \in \{0,\ldots,t\}} |u_s - (\mathsf{W}_{t,m}\boldsymbol{u})_s| = \begin{cases} 0, & t - m \le s \le t \\ |u_s|, & s < t - m \end{cases}$$

and therefore, by the monotonicity of $\boldsymbol{w}$ and the choice of $m$,

$$\max_{s \in \{0,\ldots,t\}} w_{t-s}|u_s - (\mathsf{W}\boldsymbol{u}_{t,m})_s| = \max_{s < t-m} w_{t-s}|u_s| \le w_m \|\boldsymbol{u}\|_\infty < \delta,$$

which implies that $|(\mathsf{F}\boldsymbol{u})_t - (\mathsf{FW}_{t,m}\boldsymbol{u})_t| < \varepsilon$. Consequently, $m_{\mathsf{F}}^*(\varepsilon) \le m$. Moreover, since the elements of $\boldsymbol{w}$ take values in $(0, 1]$, it follows from definitions that, for any $\boldsymbol{u}, \boldsymbol{v} \in \mathcal{M}(R)$ and any $t$,

$$\|\boldsymbol{u}_{0:t} - \boldsymbol{v}_{0:t}\|_\infty < \delta \quad \Longrightarrow \quad \max_{s \in \{0,\ldots,t\}} w_{t-s}|u_s - v_s| < \delta \quad \Longrightarrow \quad |(\mathsf{F}\boldsymbol{u})_t - (\mathsf{F}\boldsymbol{v})_t| \le \alpha_{\boldsymbol{w},\mathsf{F}}(\delta).$$

This establishes (7). $\qquad\square$

*Proof of Proposition 4.2.* The family of mappings $\varphi_{s,t}^{\boldsymbol{u}}(\cdot)$ has the following *semiflow property*: for any input $\boldsymbol{u}$ and any $0 \le r \le s \le t$,

$$\varphi_{r,t}^{\boldsymbol{u}}(\xi) = \varphi_{s,t}^{\boldsymbol{u}}(\varphi_{r,s}^{\boldsymbol{u}}(\xi)). \tag{A.2}$$

By telescoping and by the semiflow property (A.2), we have

$$\varphi_{0,t}^{\boldsymbol{u}}(\xi) - \varphi_{0,t}^{\tilde{\boldsymbol{u}}}(\xi) = \sum_{s=0}^{t-1} \left( \varphi_{s,t}^{\boldsymbol{u}}(\varphi_{0,s}^{\tilde{\boldsymbol{u}}}(\xi)) - \varphi_{s+1,t}^{\boldsymbol{u}}(\varphi_{0,s+1}^{\tilde{\boldsymbol{u}}}(\xi)) \right)$$
$$= \sum_{s=0}^{t-1} \left( \varphi_{s+1,t}^{\boldsymbol{u}}(\varphi_{s,s+1}^{\boldsymbol{u}}(\varphi_{0,s}^{\tilde{\boldsymbol{u}}}(\xi))) - \varphi_{s+1,t}^{\boldsymbol{u}}(\varphi_{0,s+1}^{\tilde{\boldsymbol{u}}}(\xi)) \right). \tag{A.3}$$

Using the fact that $\varphi_{s,s+1}^{\boldsymbol{u}}(\varphi_{0,s}^{\tilde{\boldsymbol{u}}}(\xi)) = \varphi_{s,s+1}^{\boldsymbol{u}}(f(\varphi_{0,s}^{\tilde{\boldsymbol{u}}}(\xi), u_s))$ and the stability property (9),

$$\left\| \varphi_{s+1,t}^{\boldsymbol{u}}(\varphi_{s,s+1}^{\boldsymbol{u}}(\varphi_{0,s}^{\tilde{\boldsymbol{u}}}(\xi))) - \varphi_{s+1,t}^{\boldsymbol{u}}(\varphi_{0,s+1}^{\tilde{\boldsymbol{u}}}(\xi)) \right\| \le \beta \left( \|f(\tilde{x}_s, u_s) - f(\tilde{x}_s, \tilde{u}_s)\|, t - s - 1 \right).$$

Substituting this into (A.3), we get (10). $\qquad\square$

*Proof of Proposition 4.14.* Since the matrix $A$ is Schur, the function

$$g(r) := \sup_{z \in \mathbb{T}} |G(rz)| = \|G(r \cdot)\|_{\mathcal{H}_\infty(\mathbb{T})}, \qquad r > \rho(A)$$

is continuous. In particular, there exists some $r_0 \in (\rho(A), 1)$, such that $g(r_0) < g(1) < \gamma^{-1}$. Consequently, the rational function

$$H(z) := \gamma G(r_0 z) = \frac{\gamma C}{r_0} \left( zI_n - \frac{A}{r_0} \right)^{-1} B$$

is well-defined for all $z \in \mathbb{C}$ with $|z| \geq r_0$, and we have the following:

- $\frac{A}{r_0}$ is a Schur matrix;

- the pair $(\frac{A}{r_0}, B)$ is controllable;

- the pair $(\frac{A}{r_0}, \frac{\gamma C}{r_0})$ is observable;

- $\|H\|_{\mathcal{H}_\infty(\mathbb{T})} < 1$.

Then, by the Discrete-Time Bounded-Real Lemma [Vaidyanathan, 1985], there exist real matrices $L, W$ and a symmetric positive definite matrix $P \in \mathbb{R}^{n \times n}$, such that

$$A^\top P A + \gamma^2 C^\top C + r_0^2 L^\top L = r_0^2 P \tag{A.4a}$$

$$B^\top P B + W^\top W = I_n \tag{A.4b}$$

$$A^\top P B + r_0 L^\top W = r_0 I_n. \tag{A.4c}$$

From (A.4), for any $\theta \in \mathbb{R}$ we have

$$(A - \theta BC)^\top P(A - \theta BC) - r_0^2 P$$
$$= A^\top P A - \theta(C^\top B^\top P A + A^\top P BC) + \theta^2 C^\top B^\top P BC - r_0^2 P$$
$$= (\theta^2 - \gamma^2) C^\top C - (r_0 L - \theta WC)^\top (r_0 L - \theta WC).$$

Let $\mu := r_0^2$. Then, since $\gamma^2 \geq \theta^2$ for all $\theta \in [a, b]$, it follows that

$$(A - \theta BC)^\top P(A - \theta BC) - \mu P \preceq 0, \qquad a \leq \theta \leq b.$$

Since

$$\frac{\partial}{\partial x} f(x, u) = \frac{\partial}{\partial x} \left( Ax + B\psi(u - Cx) \right) = A - \psi'(u - Cx)BC$$

and $\psi'(u - Cx) \in [a, b]$ for all $x$ and $u$, the proposition is proved. $\qquad\square$