[Reviews · NeurIPS 2019]

Reviewer 1



This paper brings a powerful dynamical systems perspective to bear on recurrent models. The universal approximation theory and results on incrementally stable models offer theoretical support for why Bai et al. 2018 and others can outperform recurrent models with feedforward/TCN architectures, and the approximation results nicely generalize those of Miller and Hardt 2019. In particular, the approximation theorem avoids relying on a state-space representation, and the Theorem 4.1 showing recurrent models can have approximately finite memory goes through without the strong exponential contractivity assumption. The paper is clearly written, and I checked most of the proofs for correctness. In addition to the results themselves, the paper offers a nice bridge between recurrent models and powerful results, tools, and definitions in dynamical systems and controls that hopefully inspire more interplay between the two areas in the future. Since the results in the paper are quite general, I would have appreciated instantiating them with a few specific parameterizations. For instance, the paper derives the Miller/Hardt result as a special case of a more powerful result. Are there nice examples where models fail the more restrictive stability condition, but satisfy incremental stability and show approximately finite memory? Finally, is the exponential dependence on depth in theorem 3.1 inevitable or an artifact of the construction via Hanin and Selke 2018? Typos: Equation 17: f(\xi), u should be f(\xi, u) and similarly later in the proof. After rebuttal: Thanks for the authors for their response. I remain a fan of this paper. The specific example given showing a separation between the contractivity condition and the conditions studied in the paper is interesting, and the camera-ready version might benefit from including this and other examples, or at least linking to them the appendix for the interested reader.

Reviewer 2



Overall I thought the paper was quite clear: the definitions were precise and given in a timely manner, and the theorems followed in a reasonable order with plenty of motivating / explanatory remarks. I am less sure about how deep / novel the results are however. On Theorem 3.1 - I think this is a relatively trivial application of [Hanin and Sellke, 2018], and so does not contribute much to the body of ML knowledge? The result of Hanin and Sellke says that "we can well-approximate functions on m variables with a net of width m+1 and some depth" - and Theorem 3.1 says that "if you have a function on an infinite number of variables (i/o map) that only uses approximately local context of m variables, then you can well-approximate it by a (convolutional) net of width m+1 and some depth", which follows in a pretty straightforward manner. Some other remarks: Definition of mF*(0): for this to be well-defined, it should be "<= epsilon", not "< epsilon", in defition 2.1 (equation 1). I suggest sharing same number counter for propositions, theorems, definitions, etc. Otherwise there is a remark, theorem, assumption, proposition, and definition all with the label 3.1, which is slightly confusing. On definition of time-invariant: there is an annoying condition here on "boundary effects" at t=0. In particular it requires that F(0, x_1, x_2, ...) = 0, F(x_1, x_2, ..) (also: does k>1 follow from k=1?), but recurrent systems (as defined at the beginning of section 4) only satisfy this if they ignore initial sequences of zero, i.e., with the extra conditions that f(\xi, 0) = 0 and g(\xi) = \xi where \xi \in R^n is the initial state. Therefore the time-invariant defition, or the conditions on recurrent models, need to be adjusted.

Reviewer 3



Summary of main ideas: This paper consists of three theorems. Theorem 3.1 states that a ReLU TCN can $\epsilon$-approximate (in $L^\infty$ sense) any causal and time-invariant i/o map F with (Assumption 3.1) approximately finite memory and (Assumption 3.2) an additional continuity condition. Theorem 4.1 states that a state-space model with (Assumption 4.1) sufficiently smooth transition/output maps, (Assumption 4.2) compact state spaces, and (Assumption 4.3) uniformly asymptotically incrementally stability'' generates a causal and time-invariant i/o map F. Theorem 4.2 (and Corollary 4.1) provides detailed estimates of the memory and continuity against the norm $R$ of inputs, the Lipschitz constants $L_f$ (of transition model) and $L_g$ (of observation model), under some assumptions including Demidovich criterion,'' which contains the conditions assumed in [Miller and Hardt, 2019] as a special case. Originality: High. The theorems are new and consistent. Quality: High. The most technical part (Theorem 3.1) seems to be imported from [Hanin and Sellke, 2018]; but the total message goes beyond the previous work. Clarity: High. The writing is good. Significance: Medium. Since this study is heavily motivated by [Miler and Hardt, 2019], I would raise my score if the authors could answer the learnability question: What class of i/o maps a TCN can learn during gradient descent training? After rebuttal: I appreciate the authors' reply. I agree that the authors are right that the learnability arguments need explicit parameterizations, but surely it is out-of-the-scope in this study. I will raise my score because I am also moved by the R1's strong phrase "I expect to see follow-up work exploiting these definitions."

[Author Response · NeurIPS 2019]

**General response:** We would like to thank the reviewers for their comments. We will incorporate all of the suggestions in the final revision.

**Responses to comments of reviewer 1:**

**Comment 1:** Are there nice examples where models fail the more restrictive stability condition, but satisfy incremental stability and show approximately finite memory?

**Response 1:** Additional specific examples were not included in the paper due to lack of space, but a large class of nonlinear systems which, in general, fail the contractive stability condition of Miller and Hardt but satisfy incremental stability and approximately finite memory is linear time-invariant systems connected with a nonlinear feedback element satisfying the so-called circle criterion (see, e.g., M. Vidyasagar, *Nonlinear Systems Analysis*). This includes systems of the form $x_{t+1} = Dx_t + C\sigma(Ax_t + Bu_t)$, where $\sigma$ is a componentwise application of a Lipschitz-continuous nonlinearity. Here, $f(x, u) = Dx + C\sigma(Ax + Bu)$ is Lipschitz, but it need not be contractive. On the other hand, it can be shown that the circle criterion is sufficient for approximately finite memory and for global exponential stability, without enforcing contractivity of the state transition map.

**Comment 2:** Is the exponential dependence on depth in Theorem 3.1 inevitable or an artifact of the construction via Hanin and Selke 2018?

**Response 2:** The exponential width dependence of the depth of a minimal-width ReLU network is a consequence of Hanin and Sellke's construction. This can be seen in the last paragraph of the proof of Proposition 3 in their paper. When the output of the ReLU net is scalar, the minimal width is equal to $d + 1$, where $d$ is the input dimension, so the number of neurons is exponential in $d$. On the other hand, exponential dependence on $d$ is generally inevitable when approximating continuous functions by deep ReLU nets, as shown by D. Yarotsky (COLT 2018).

**Responses to comments of reviewer 2:**

**Comment 1:** On Theorem 3.1 - I think this is a relatively trivial application of [Hanin and Sellke, 2018] ... Have I missed something in why the result does not follow in a relatively straightforward manner from Hanin and Sellke?

**Response 1:** We agree that it is at least mildly surprising how easily this result follows from an existing result on function approximation by neural nets (modulo a careful application of causality and time-invariance to relate everything to the output of F at time $t$). However, to the best of our knowledge, all existing results on universal approximation of i/o maps (e.g., by Boyd–Chua or by Sandberg) reduce the problem to universal approximation of continuous functions on an appropriate compact set and then apply a suitable version of the Stone–Weierstrass theorem. A common drawback is that these proofs are nonconstructive. What we were after was a *quantitative* version of Stone–Weierstrass that would allow us to isolate explicitly the dependence of the depth and width of the approximating ReLU TCN on the approxiate memory length and on the modulus of continuity associated to the original i/o map F. Although Hanin and Sellke do not mention this, their result is, essentially, a quantitative formulation of the Kakutani–Krein theorem, which guarantees that any continuous real-valued function on a compact set can be approximated by a finite composition of affine maps and lattice operations. We will emphasize these points in the final version.

**Comment 2:** On definition of time-invariance.

**Response 2:** Thank you for pointing out this oversight. The correct definition of time invariance should be as follows (from Sandberg, 1991): $(FR^k\mathbf{u})_t = 0$ for $t < k$ and $(FR^k\mathbf{u})_t = (F\mathbf{u})_{t-k}$ for $t \geq k$. The recurrent model of Section 4 will be time-invariant if the initial state $\xi$ satisfies the conditions $f(\xi, 0) = \xi$ and $g(\xi) = 0$.

**Response to comments of reviewer 3:**

**Comment 1:** Since this study is heavily motivated by [Miller and Hardt, 2019], I would raise my score if the authors could answer the learnability question: What class of i/o maps a TCN can learn during gradient descent training?

**Response 1:** The work of Miller and Hardt was concerned with both approximation and learning. For the latter, they showed that any strictly contracting recurrent model can be approximately learned using gradient descent with truncated backpropagation through time. Since the original model can be learned using backpropagation through time, it is meaningful to compare the gradient descent trajectories with and without truncation. Note that one needs an explicit state-space realization in order to write down the gradient update equations. By contrast, our goal was to show that TCNs can approximate a much wider class of i/o maps with approximately finite memory. Since a TCN model applies a fixed feedforward deep ReLU net to shifted copies of the input training sequence, one can use standard gradient descent with backprop for training.

[Meta-Review · NeurIPS 2019]

The first contribution of this paper is Theorem 3.1 that shows that ReLU TCNs (temporal convolutional networks) can arbitrarily approximate any causal, time-invariant map. Moreover, it provides quantitative bounds the properties of the approximating network (context length, width and depth) in terms of the continuity parameters of the map being approximated. The second contribution is Theorem 4.1 which gives incremental stability conditions on state-space maps (or recurrent models) such that the i/o maps they induce satisfy the assumptions of Theorem 3.1. The third contribution (Theorem 4.2) is of the same kind as Theorem 4.1 but now under an exponential incremental stability condition along with a stability criterion from nonlinear systems theory called Demidovich criterion. I agree with R1 that an additional strength of the paper is in drawing connections between dynamical systems theory and recurrent models. R2 had concerns about the significance of Theorem 3.1 in light of [Hanin and Sellke, 2018]. However, in the discussions R1 pointed out that the simplicity of the proof is due to the work on the part of the authors in setting up the framework and definitions so that the conclusions follows easily. Such simplicity is deceptive and usually takes work to achieve. I agree with R1's sentiment. The authors should revise the manuscript to make the changes the reviewers have suggested. In particular, all typos etc. should be fixed before the final version is submitted.